# Ultra-Small Nanoparticles of Pd-Pt-Ni Alloy Octahedra with High Lattice Strain for Efficient Oxygen Reduction Reaction

Yuanyan Luo [1], Wenhua Lou [1], Huiyan Feng [1], Zhihang Liu [1], Qiuyan Chen [1], Guizhen Liao [1], Xiaoting Huang [1], Panagiotis Tsiakaras [2,*] and Peikang Shen [1,*]

1   Collaborative Innovation Center of Sustainable Energy Materials, School of Physical Science and Technology, Guangxi Key Laboratory of Electrochemical Energy Materials, Key Laboratory of New Processing Technology for Non-Ferrous Metal and Materials of Ministry of Education, Guangxi University, Nanning 530004, China
2   Laboratory of Alternative Energy Conversion Systems, Department of Mechanical Engineering, School of Engineering, University of Thessaly, 1 Sekeri Str., Pedion Areos, 38834 Volos, Greece
*   Correspondence: tsiak@mie.uth.gr (P.T.); pkshen@gxu.edu.cn (P.S.)

**Abstract:** The design and synthesis of ultra-small-sized Pt-based catalyst with specific effects for enhancing the oxygen reduction reaction (ORR) is an effective way to improve the utilization of Pt. Herein, Pt-Pd-Ni octahedra nanoparticles characterized by the ultra-small size of 4.71 nm were synthesized by a Pd seed-inducing-growth route. Initially, Pd nanocubes were synthesized under solvothermal conditions; subsequently, Pt-Ni was deposited in the Pd seed solution. The Pd seeds were oxidized into $Pd^{2+}$ and combined with $Pt^{2+}$ and $Ni^{2+}$ in the solution and finally formed the ternary alloy small-sized octahedra. In the synthesis process of the ultra-small Pt-Pd-Ni octahedra, Pd nanocube seed played an important role. In addition, the size of the Pt-Pd-Ni octahedra could be regulated by adjusting the concentration rate of Pt-Ni. The ultra-small Pt-Pd-Ni octahedra formation by depositing Pt-Ni with a feeding ratio of 2:1 showed good ORR activity, and the high half-wave potential was 0.933 V. In addition, the Pt-Pd-Ni octahedra showed an enhanced mass activity of 0.93 A $mg^{-1}$ $_{Pt+Pd}$ in ORR, which was 5.81 times higher than commercial Pt/C. The theoretical calculation shows that compared to Pt/C, the small-sized ternary alloy octahedra had an obvious contraction strain effect (contraction rate: 3.49%). The alloying effect affected the d-band center of the Pt negative shift. In the four-electron reaction, Pt-Pd-Ni ultra-small octahedra exhibited the lowest overpotential, resulting in the adsorption performance to become optimized. Therefore, the Pd seed-inducing-growth route provides a new idea for exploring the synthesis of small-sized nanoparticle catalysts.

**Keywords:** ultra-small sizes; Pd-Pt-Ni octahedra; Pd seed-inducing-growth route; lattice contraction rates; D-band center; oxygen reduction reaction

## 1. Introduction

Proton exchange membrane fuel cells (PEMFCs) are energy conversion devices with great development potential for automobile and mobile electronic equipment due to the fact of their facile fuel transportation and environmental benefits [1–7]. Oxygen reduction reaction (ORR) is one of the main cathode reactions and plays an integral part in PEMFCs [8]. Advanced electrocatalysts are the key to improving the kinetic of the fuel cell [4,9,10]. Among them, platinum (Pt)-based nanomaterials can remarkably reduce the activation energy of a cathodic reaction [11]. However, due to the rarity of Pt-based materials, it is necessary to explore improving Pt utilization and find the essence of the Pt-based catalytic activity [4,7,9,10,12–17].

In many current reports, two strategies are suggested for catalyst development. (1) One is to increase the number of active sites on the Pt surface. More precisely, the

amount of Pt-O bond formation can be increased by adjusting the surface/volume ratio of nanoparticles or exposing specific crystal faces [18]. This approach relies on the synthesis of different structures, such as one-dimensional nanowires, two-dimensional nanosheets, the three-dimensional structure of the nanoframe and octahedra with specific crystal planes [4,7,14,17,19]. (2) The other strategy is to improve the activity of the intrinsic active site to optimize the adsorption and dissociation of the related reactants [20]. This approach relies on the synthesis of multicomponent structures, such as alloys, overlays, or the introduction of dopants or vacancies into the structure. Specifically, alloying with Pt can be formed through strain effect and synergistic effect among different metals. The core–shell structure is the most representative [4,9,10,16,21]. However, even though these nanostructures have improved the ORR activity, there are still shortcomings. One of the most important reasons is the regulation of size; the larger catalyst still has some limitations in the reaction conditions and selectivity of the reaction mechanism [22–25]. Therefore, the further development of the synthesis of small nanoparticles is still important for Pt-based catalysis.

Nanoparticles with a size of less than 5 nm have been widely studied and discussed because of their large specific surface area, which can fully expose more active sites [26]. Moreover, ultra-small nanoparticles have more under-coordinated atoms on kinks, edges, steps, or open surfaces (111) and (110), which directly optimizes ORR activity [23,24,27,28]. At present, the high-temperature annealing method and template load method are the main methods for the synthesis of small sizes [29,30]. Jia et al. synthesized an ordered structure of Pt-Co/HSC and $PtCo_3$/HSC NPS deployed by high-temperature annealing, which regulates the structure of the intermetallic compounds at the atomic level, shows a breakthrough in the field of catalysts, but the nanoparticles will agglomerate at such high temperatures, causing a decrease in the specific surface [31]. Stephens et al. pointed out the difficulty in eliminating the interference of other factors (such as composition, alloying degree, and annealing environment), by using the high-temperature annealing method [25,28,32]. Another strategy is to load a template onto a nanoparticle, such as Wu synthesized ORR catalysts by integrating PGM NPs and the FeN4 site-rich Fe–N–C carbon denoted as Pt/FeN4–C or PtCo/FeN4–C, but the ORR activity is not very high (0.71 A mg $^{-1}_{Pt}$) [33]. Therefore, based on the above empirical theory, a more simple and safe strategy was designed to synthesize ultra-small and super-strong alloying catalysts [34].

In this work, we report a Pd seed inducing-growth route for preparing small-sized ternary alloy octahedra (Pd-Pt Ni octa1, 4.71 nm), basic deposition Pt-Ni to induce Pd seed for lattice distortion and structure remodeling strategy [4]. Concretely, the small-sized octahedra were obtained by the structural remodeling of Pd nanocubes seed after Pt-Ni deposition and the alloying effect with $Pt^{2+}$, $Ni^{2+}$, and $Pd^{2+}$ [35]. In the synthesized process of ultra-small Pt-Pd-Ni octahedra, the Pd nanocubes seed formed is the key intermediate [20]. The size of the Pt-Pd-Ni octahedra can be regulated by adjusting the feeding ratio of Pt-Ni. The ultra-small Pt-Pd-Ni octahedra formation, by depositing Pt-Ni with a feeding ratio of 2:1, shows excellent ORR activity, whose high half-wave potential (0.933 V) is higher than Pt/C (0.897 V). In addition, ultra-small Pt-Pd-Ni octahedra showed strong strain effects. The theoretical calculation shows that compared with Pt/C, the ultra-small size ternary alloy octahedra has an obvious contraction strain effect (the compression ratio is found to be 3.49%). The alloying effect affects the d-band center of Pt negatively shift. In the four-electron reaction, Pt-Pd-Ni ultra-small octahedra exhibit the lowest overpotential, resulting in the adsorption performance being optimized. Therefore, the Pd seed-inducing-growth route will provide a new idea for exploring the synthesis of alloy catalysts with small-sized nanoparticles [5,15,25,36].

## 2. Results and Discussion

### 2.1. Physicochemical Characterization

The growth progress of the Pd-Pt-Ni Octahedra is illustrated in Scheme 1.

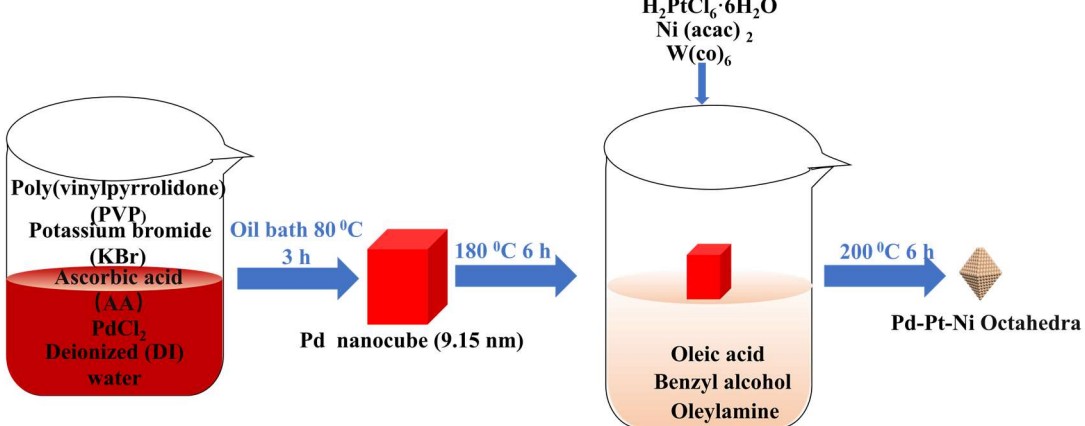

**Scheme 1.** Schematic representation of the synthesis procedure via deposition-induced Pd seed lattice distortion and structural change strategy.

The Pd nanocubes (Pd NCs) were synthesized using PVP as a capping agent, DI water as a solvent, KBr as a surfactant, and AA as a reducing agent (Figure 1a,b). The particle sizes were approximately 9.17 nm. Subsequently, the centrifuged Pd NC seeds were vaporized by stirring at a high temperature to remove the remaining water. When heated to a higher temperature, ligand exchange will occur, and the PVP ligand will be replaced by the OAM ligand (-NH2) with a stronger coordination ability to become OAM-Pd. Then, the Pd NC solution was converted to an oleylamine (OAM) system [37]. Subsequently, the precursor material we selected was the $H_2PtCl_6$ $6H_2O$, which has a fast reduction rate. The feeding ratio Pt:Pd:Ni was 2:2:1. When the transition metal was introduced, it changed the electronic structure of the Pt atoms with lattice to minimize the surface energy and enhance the ORR. In addition, Pd has a similar electronic structure and lattice constant as Pt, and Pd is also affected by Ni in ternary alloys. Therefore, the displacement reaction between Pd and Pt occurs spontaneously under the condition of OAM and for a longer reaction time.

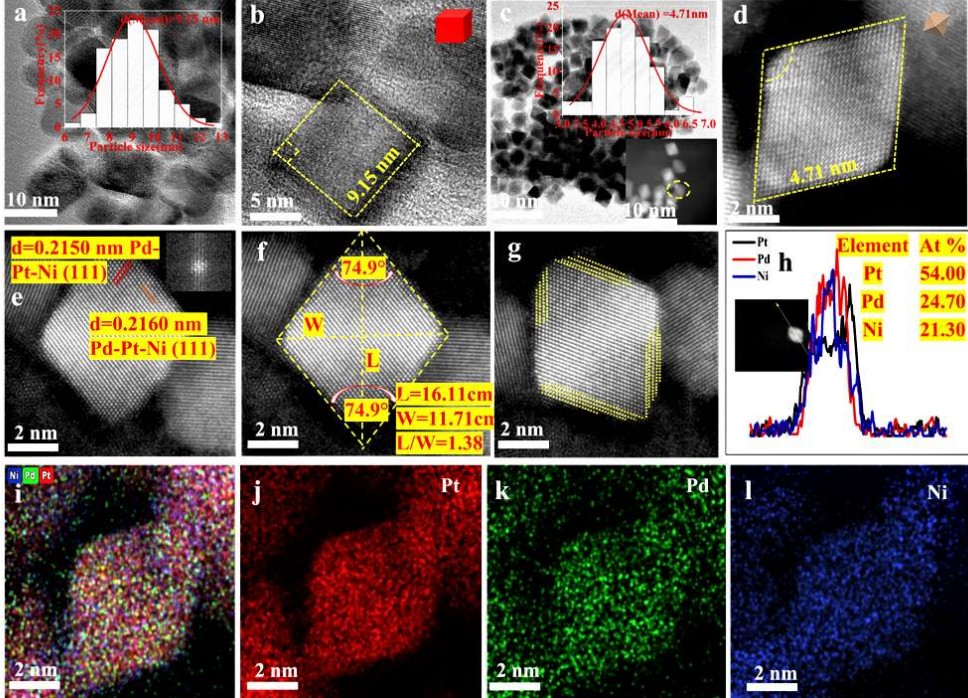

**Figure 1.** (**a–k**) TEM images of Pd nanocube (Pd NCs) seed and ultra-small octahedra (Pd-Pt-Ni Octa1): (**a**) TEM image of Pd NC seed with particle size distribution; (**b**) TEM image of a single Pd NC;

(**c**) TEM image of Pd-Pt-Ni Octa1 with particle size distribution; (**d**) TEM image of a single Pd-Pt-Ni Octa1; (**e**) high-magnification TEM image of Pd-Pt-Ni Octa1 and calibration of atomic spacing and lattice constants; (**f**,**g**) high-magnification TEM image of Pd-Pt-Ni Octa1 and calibration of angles, steps, and kink; (**h**) TEM image and line scanning analysis; (**i**–**l**) HAADF-STEM images with EDS mapping of a single Pd-Pt-Ni Octa1.

Therefore, the small-sized octahedra (Pd-Pt-Ni Octa1) were obtained using Pd NCs as the substrate and Pt -Ni as the precursor (Figures 1c,d and S1). The particle sizes of the Pd-Pt-Ni Octa1 were approximately 4.71 nm, and the feeding ratio of Pt-Ni was 2:1. In the process of synthesis, BA was used as a dispersant, and OAM and OA were used as weak reducing agents [38,39]. Figure 1e shows high-resolution HRTEM images of Pd-Pt-Ni Octa1 observed along the (011) direction. This demonstrates that the nanocrystals were octahedral structures with distinct shapes and densely packed atoms, meaning that the previous Pd nanocube seed structure changed after the deposition of Pt-Ni. It can also be observed that the lattice spacing of 0.2150 and 0.2160 nm in the nucleus corresponded to the Pt-Pd-Ni (111) crystal. The lattice mismatch rate between the Pd-Pt-Ni (111) crystal plane and Pd/C (0.2282 nm) crystal plane was 5.38%, which indicates that the deposition of the Pt-Ni caused lattice distortion of the Pd nanocubes seeds, which promoted the lattice mismatch rate between the newly crystal surface and previous Pd (111). Moreover, the lattice mismatch rate between the Pd-Pt-Ni (111) crystal plane and Pt/C (0.2265 nm) crystal plane was 4.63%; it also shows that the formed homogeneous Pd-Pt-Ni (111) crystal surface had an obvious contraction strain effect [40]. The lattice contraction indicates that $Pt^{2+}$, $Ni^{2+}$, and $Pd^{2+}$ generate the alloyed crystal surface Pt-Pd-Ni (111) in the deposition reaction. Related studies have shown that the lattice contraction rate is a symbol of the activity and durability of the ORR catalysts [13].

In the deposition reaction process, the number of Ni atoms was insufficient. At this point, the formation rate of Pt-Ni-Pd bonds was delayed, while the length of the Pt-Pt bond was unstable and shortened. This leads to the decrease in the $d_{ij}$ parameter in the formula:

$$V_i = 7.62 \sum_{j=1}^{CN} \frac{[r_d^{(i)} r_d^{(j)}]^{3/2}}{d_{ij}^s}$$

Herein, $V_i$ represents the interatomic matrix element describing the surface atom i and its adjacent atoms. According to the tight binding theory, it is proportional to the d-band center width ($W_d$) and monotonically related to the d-band center ($\varepsilon_d$) [31]. The $r_d^{(i)}$ represents the characteristic radius related to the spatial range of the d orbital of the ith atom, while $C_n$ represents the generalized coordination number [31]. The distance between the ith atom and the jth atom is expressed as $d_{ij}$ [31]. The shortening of $d_{ij}$ leads to the widening of the energy band and the downward shift of the center of the material's d-band, thus reducing the binding energy of O, H, OH, and other simple adsorbents and, thus, the ORR activity is increased [14,31].

Additionally, the small particle sizes of Pd-Pt-Ni Octa1 (4.71 nm) indicated the presence of kinks on the surface of the nanocrystals (Figure 1f,g). It was clearly shown that Pd-Pt-Ni Octa1 had assumed a rhomboid profile. The length–width rates of the Pd-Pt-Ni Octa1 were approximately 1.38 under electron beam irradiation, and the vertical angle on both sides was approximately 74.9°, which is quite different from a standard octahedron (usually 70°) [7], suggesting the presence of kinks on the surface of Pd-Pt-Ni Octa1. The distribution of the Pt, Pd, and Ni of Pd-Pt-Ni Octa1 was obtained by EDX mapping (Figure 1i–l). The nanoparticles formed had a strong alloy effect and were composed of Pt, Pd, and Ni; these elements all showed an octahedral morphology and a ternary alloy phase. Figure 1h shows low-resolution TEM images and a line profile analysis of a single particle, which indicates that ultra-small Pd-Pt-Ni Octa1 was a super alloying nanoparticle. The atomic ratio of Pt:Pd:Ni was approximately 5:2:2, and the number of Pt atoms was the largest, indicating

that the displacement reaction between Pt and Pd occurred when Pt-Ni was deposited on Pd seed, leading to the decrease in Pd atoms in the formed octahedra structure and part of the Pd atoms precipitated into the solution. At the same time, the Pt element on the surface increased obviously, which was beneficial to the ORR.

To explore the Pd-Pt-Ni Octa1 formation process and the effect of the deposition time on the Pd seeds, the relative structure was analyzed at different reaction times. In the initial stage of the reaction at 3 h (Figure 2a), it was obvious that the Pd NCs first became a triangle structure and then slowly grew into an octahedra structure. When the reaction time was increased at 6 h (Figure 2b), an ultra-small octahedra structure with a regular and good shape was synthesized.

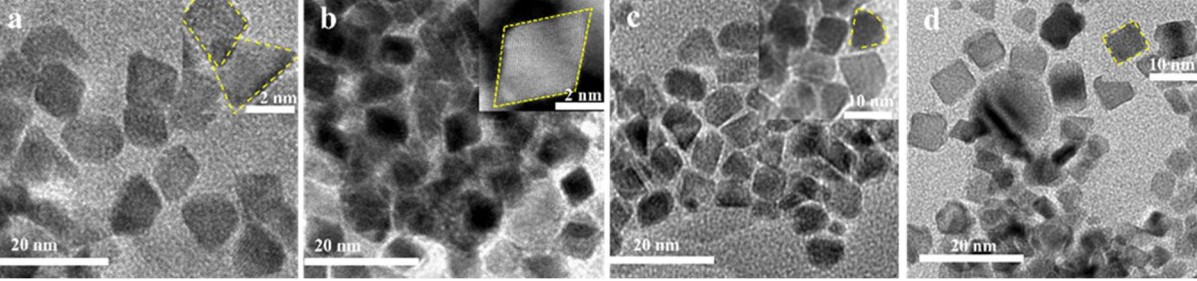

**Figure 2.** (**a–d**). TEM images of Pd-Pt-Ni ternary alloy catalyst obtained at different deposition times: (**a**) 3; (**b**) 6; (**c**) 9; (**d**) 12 h.

When the reaction time was increased at 9 h (Figure 2c), it was found that the small octahedral structure at 6 h had begun to gradually smooth the edges and transformed into a larger nanocube structure. When the reaction time was increased at 12 h (Figure 2d), the larger concave cubes were slowly being synthesized. These exploratory experiments indicate that the deposition time had a certain influence on the synthesis of ultra-small nanoparticle catalysts (less than 5 nm). The Pt-Ni deposition could induce structural changes in the Pd nanocube seeds, but when the deposition time was too long, it was not conducive to the formation of small-sized nanoparticle catalysts.

To further explore the formation progress of the ultra-small Pd-Pt-Ni Octa1 in detail, the Pt-Ni nanoparticle catalyst, Pt-Pd nanospheres, Pd-Ni nanoparticle catalyst, and Pt-Pd-Ni alloy catalyst by the one-step solvothermal method were analyzed in the same reaction state. As shown in Figure 3a,b, the Pt-Pd-Ni alloy catalyst was synthesized by the one-step solvothermal method, in which the structure was large with heterogeneous nanocubes with many growing dendrites. The one-step solvothermal method cannot synthesize octahedra of a small size, which shows the importance of Pd seeds in the whole synthesis process of small-sized nanoparticles. Figure 3c shows an image of the Pt-Ni bimetallic alloy catalyst with an irregular shape synthesized without Pd seed. It can be observed that Pt-Ni octahedra with the size of 7.02 nm were much larger than Pt-Pd-Ni octa1 (4.71 nm). Therefore, it was necessary to use Pd nanocube seed as the substrate to synthesize small-sized nano-catalyst. Figure 3d shows an image of the Pd-Ni bimetallic alloy catalyst; it can seem that there were many unregular structures, such as huge octahedra and the agglomeration of nanoparticles. The huge octahedra were the structure of $W(CO)_6$, which does not participate in the reaction process. As shown in Figure 3e,f, a nanosphere catalyst the size of 11 nm was synthesized using Pd nanocube seed as substrate and $Pt^{2+}$ as the precursor. It was slightly larger than the Pd NCs; the atomic ratio of the Pt:Pd was approximately 1:2. Because the deposition rate of Pt is higher than Pt's diffusion rate, it finally stayed on the surface, resulting in a spontaneous **displacement** reaction between Pt and Pd. Therefore, the size of the nanosphere catalyst increased. The above phenomenon indicates that the shape of the Pd seeds was affected by the deposition of Pt-Ni. The synergistic effect between Pt-Ni directly resulted in a significant structural change in the Pt

nanocube seeds, forming ultra-small-sized octahedra. The research experiments proved the effectiveness of the Pd seed-inducing-growth route.

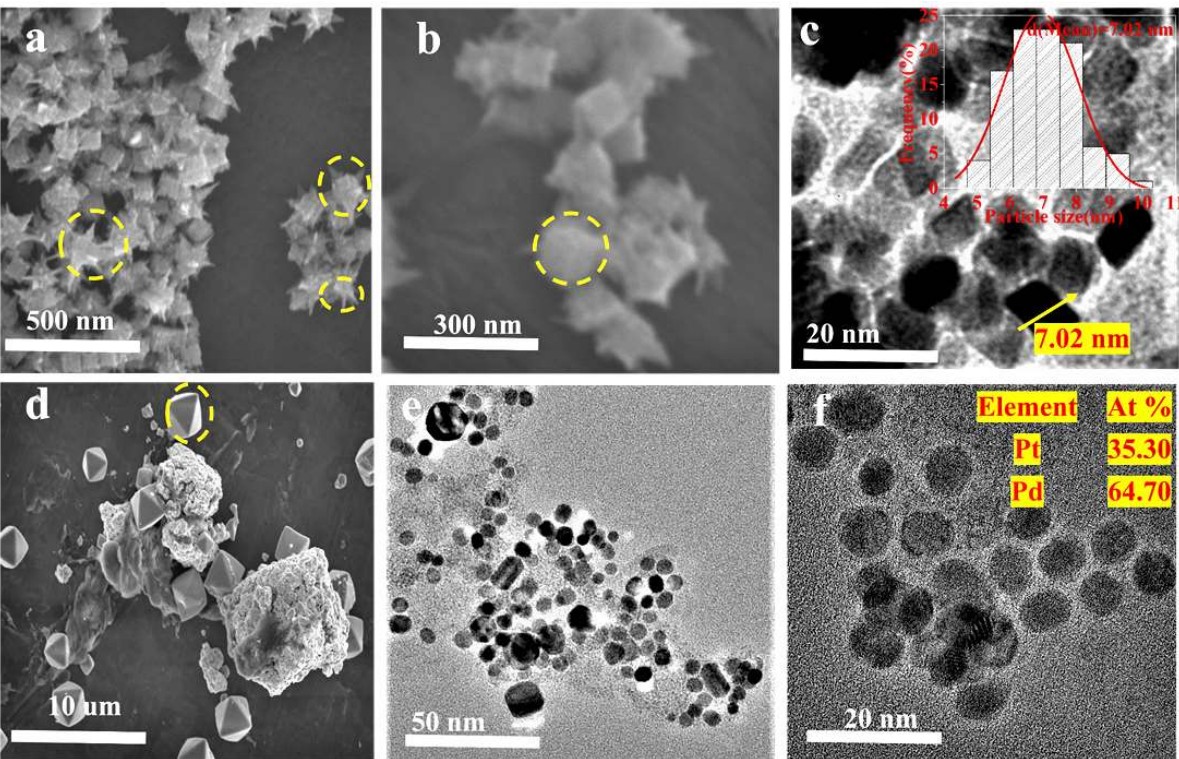

**Figure 3.** (**a**–**f**) SEM and TEM images of Pt-Ni nanoparticle catalyst, Pt-Pd nanospheres, Pd-Ni nanoparticle catalyst, and Pt-Pd-Ni alloy catalyst by the one-step solvothermal method: (**a**,**b**) Pt-Pd-Ni alloy catalyst synthesized by the one-step solvothermal method; (**c**) Pt-Ni bimetallic alloy catalyst; (**d**) bimetallic alloy catalyst synthesized by Pd seed and Ni; (**e**,**f**) bimetallic alloy catalyst synthesized by Pd seed and Pt.

Different Pt-Ni feeding ratios had different effects on the Pd seeds in the deposition reaction. The Pd-Pt-Ni octahedra2 (Pd-Pt-Ni Octa2) with a particle size of 7.37 nm were synthesized at a feeding ratio Pt:Ni of 1:1 (Figures 4a and S2), while the Pd-Pt-Ni Octa3 with a particle size of 8.51 nm (Figures 4c and S3) were synthesized when the feeding ratio of Pt-Ni was 1:2.

Therefore, it is obvious that the different Pt-Ni concentrations caused different structural changes in the Pd seeds, which is reflected in the different sizes. Furthermore, the lattice mismatch rate between Pd-Pt-Ni Octa2 with Pd/C was approximately 2.88%, and the lattice contraction rate with Pt/C was approximately 1.98% (Figure 4b). The lattice mismatch rate of Pd-Pt-Ni Octa3 with Pd/C was approximately 2.15%, and the lattice contraction rate with Pt/C was approximately 1.28% (Figure 4d). The synthesis of the three octahedra with different sizes and the increase in the lattice mismatch rate of the Pd-Pt-Ni (111) crystal and Pd (111) crystal proved the feasibility of deposition-induced Pd seeds as a lattice distortion and structural remodeling strategy.

The XRD patterns of Pd-Pt-Ni/Octa1, Pd-Pt-Ni/Octa2, and Pd-Pt-Ni/Octa3 are displayed in Figure 5a. As can be seen, their peak values corresponded to the (111), (200), (222), and (311) crystal planes of the Pt-Pd-Ni alloy. In particular, the catalysts with different particle sizes exhibited obvious positive shifts compared with the standard peak positions of the Pt/C. Meanwhile, for the small-sized Pd-Pt-Ni Octa1 (4.71 nm), the shift was larger, and the small shift of the (111) crystal surface indicates that the interaction between Pt and other atoms was more significant in the nanocrystals. In addition, the XRD patterns also

show that the octahedral alloy of Pd-Pt-Ni Octa1, Pd-Pt-Ni Octa2, and Pd-Pt-Ni Octa3 had a good effect, and there was no interference of other miscellaneous peaks.

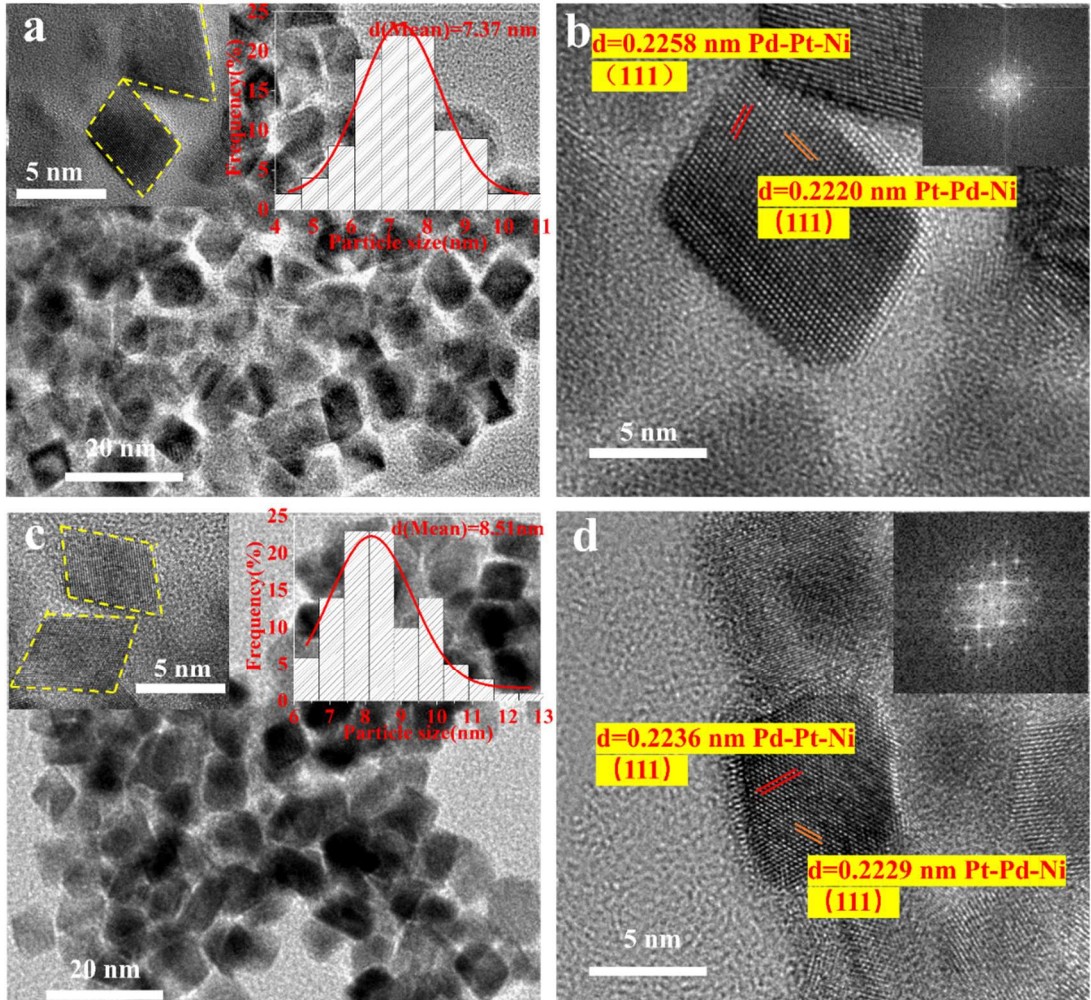

**Figure 4.** (**a**–**d**) TEM images of Pd-Pt-Ni octahedra2 (Pd-Pt-Ni Octa2) and Pd-Pt-Ni octahedra3 (Pd-Pt-Ni Octa3) obtained using various feeding ratios of Pt-Ni: (**a**) TEM images of Pd-Pt-Ni Octa2 and feeding ratios of Pt-Ni of 1:1; (**b**) HRTEM images of a single Pd-Pt-Ni Octa2 and calibration of the atomic spacing and lattice constants; (**c**) TEM images of Pd-Pt-Ni Octa3 and feeding ratios of Pt-Ni of 2:1; (**d**) HRTEM images of Pd-Pt-Ni Octa3 and calibration of the atomic spacing and lattice constants.

The chemical composition and valence states of this particle size were obtained through X-ray photoelectron spectroscopy (XPS) spectra. We can observe from Figure 5b–d that all of the samples of Pd-Pt-Ni Octa1, Pd-Pt-Ni Octa2, and Pd-Pt-Ni Octa3 presented two groups of characteristic peaks of Pt, Pd, and Ni. Compared with the standard Pt $4f_{7/2}$ (71.00 eV) and Pd $3d_{5/2}$ (336.7 eV), the binding energy of the Pd-Pt-Ni Octa1, Pd-Pt-Ni Octa2, and Pd-Pt-Ni Octa3 exhibited a positive shift, demonstrating the improvement in the electronic structure of Pt and Pd. Therefore, it can be seen that during the synthesis of the octahedra, Ni was the main dopant, and the charge transfer mainly occurred in the displacement reaction between Pt and Pd.

The increase in the binding energy can be attributed to the synergistic and alloying effects. In addition, the Pt-Ni deposition reaction resulted in the lattice change in the Pd. Comparing the XPS data listed in Table S1, the content of Pt atoms in Pd-Pt-Ni Octa1 and Pd-Pt-Ni Octa2 was maintained in a relatively stable range, but the content of Pt atoms in Pd-Pt-Ni Octa3 was rare. Among them, Pd-Pt-Ni Octa1 had the highest content of Pt

atoms, which indicates that the surface of the ultra-small octahedral was rich in Pt, which was beneficial for improving the ORR activity.

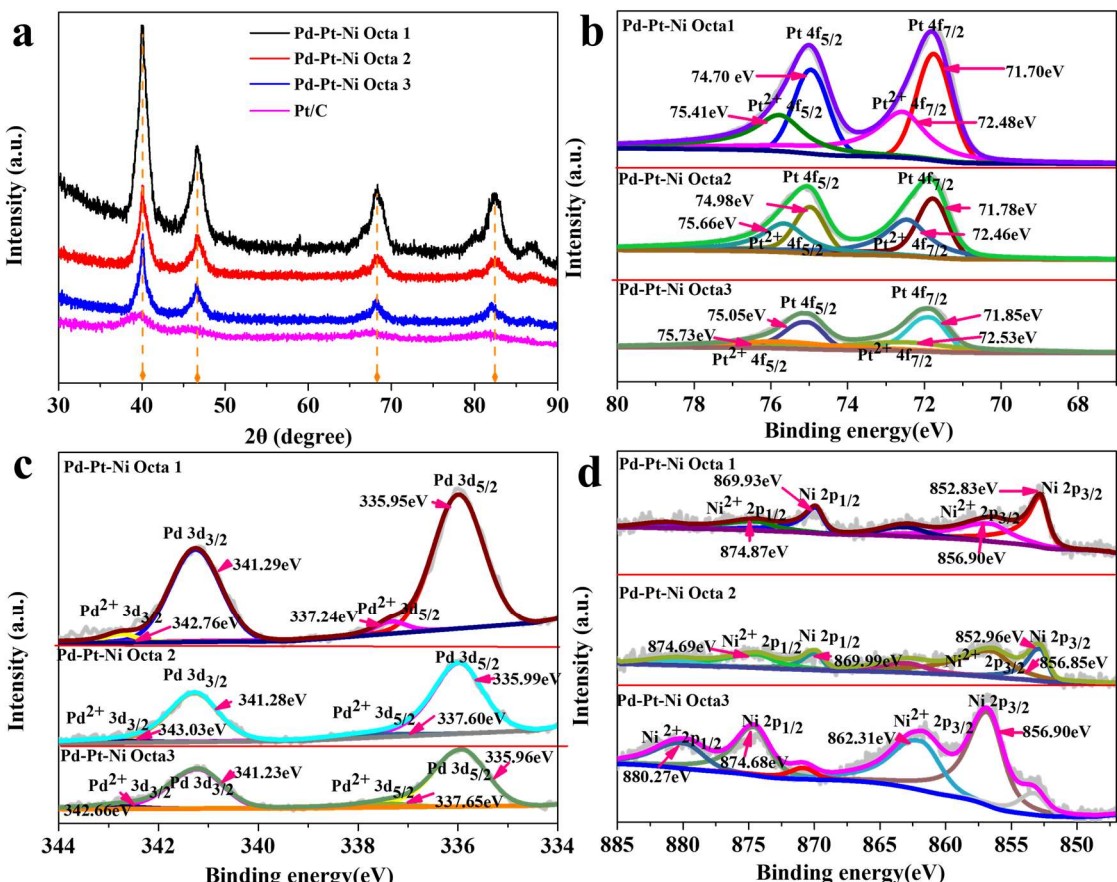

**Figure 5.** (**a–d**) XRD and XPS patterns of octahedra obtained by different Ni concentration ratios: (**a**) XRD patterns;. (**b**) Pt 4f spectra of Pd-Pt-Ni Octa1, Pd-Pt-Ni Octa2, and Pd-Pt-Ni Octa3; (**c**) Pd 3d spectra of Pd-Pt-Ni Octa1, Pd-Pt-Ni Octa2, and Pd-Pt-Ni Octa3; (**d**) Ni 2p spectra of Pd-Pt-Ni Octa1, Pd-Pt-Ni Octa2, and Pd-Pt-Ni Octa3.

*2.2. Oxygen Reduction Reaction Performance*

The different concentrations of Pt-Ni deposition had different effects on the Pd seeds, thus forming octahedral structures with different sizes and different lattice contraction rates, which will affect their ORR activity to a certain extent. The ORR activity of the catalysts was determined in the electrolytic solution of 0.1 M $HClO_4$.

Figure 6a demonstrates their cyclic voltammetry (CV) curves at a potential sweep speed of 0.05 V/s in 0.1 M $HClO_4$ solution at $N_2$ saturation. At the scanning rate of 0.05 V/s, the characteristic peak of the Pt adsorption and desorption, the peak of hydrogen and adsorption and desorption and the peak of oxygen can could be identified. The electrochemical active surface area (ECSA) (Figure 6e) of Pd-Pt-Ni Octa1, Pd-Pt-Ni Octa2, and Pd-Pt-Ni Octa3 were 103, 75, and 74 $m^2\,g^{-1}$, respectively, according to the hydrogen adsorption charge in the CV curves. The results showed that Pd-Pt-Ni Octa1 with small particle sizes less than 5 nm showed a higher specific surface area and more catalytic exposed atoms.

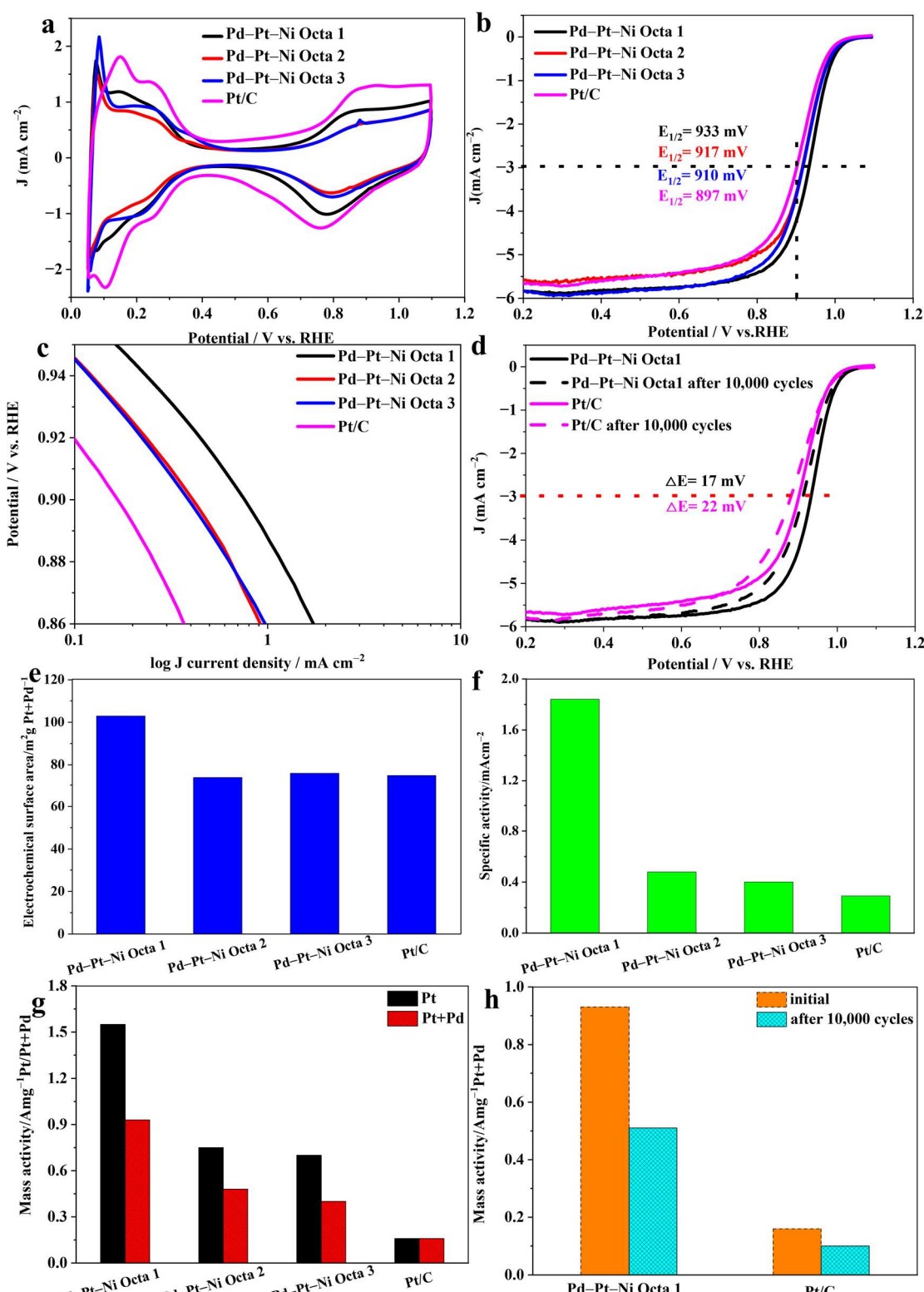

**Figure 6.** Electrochemical ORR properties of the Pd-Pt-Ni Octa1, Pd-Pt-Ni Octa2, Pd-Pt-Ni Octa3, and Pt/C: (**a**) CV curves in 0.1 M HClO$_4$; (**b**) LSV polarization curves recorded at 10 mV s$^{-1}$; (**c**) Tafel slopes; (**d**) LSV polarization curves of Pd-Pt-Ni Octa1 and commercial Pt/C before and after scanning 10,000 cycles; (**e**) electrochemical surface area (ECSA) comparison; (**f**) specific activity (SA) comparison of the four catalysts; (**g**) mass activity (MA) comparison; (**h**) performance comparison with commercial Pt/C after scanning 10,000 cycles.

When the size increased, the specific surface area became smaller, which was consistent with the results of the electron microscope characterization. Then, the ORR polarization curves of Pd-Pt-Ni Octa1, Pd-Pt-Ni Octa2, and Pd-Pt-Ni Octa3 were measured at a potential scanning rate of 10 mV/s in $O_2$ saturated in the solution of 0.1 M $HClO_4$ (Figure 6b).

According to the polarization curve analysis, Pd-Pt-Ni Octa1, Pd-Pt-Ni Octa2, and Pd-Pt-Ni Octa3 showed better half-wave potentials, which were 0.933, 0.917, and 0.91 V, respectively. Compared with Pt/C electrocatalysts (0.897 V), the ORR performance was significantly enhanced, especially the ultra-small Pd-Pt-Ni Octa1. Figure 6c was an analysis of the Tafel slope related to its dynamic current and $M_{Pt+Pd}$ load. It can be fully proved that the slopes of Pd-Pt-Ni Octa1, Pd-Pt-Ni Octa2, and Pd-Pt-Ni Octa3 were significantly lower than pure Pt/C. This shows that Pd-Pt-Ni Octa1, Pd-Pt-Ni Octa2, and Pd-Pt-Ni Octa3 had higher electron transfer rates than Pt/C, and Pd-Pt-Ni Octa1 had the fastest electron transfer rate. This indicates that the ultra-small Pd-Pt-Ni Octa1 had a faster kinetic rate compared with Pd-Pt-Ni Octa2 and Pd-Pt-Ni Octa3, which causes their peak potential to be more negative and to have a lower overpotential and, thus, have better ORR activity. In addition, the mass activity (MA) (Figure 6g) and specific activity (SA) (Figure 6f) were calculated according to the CV and ORR polarization curves. The MA of Pd-Pt-Ni Octa1, Pd-Pt-Ni Octa2, and Pd-Pt-Ni Octa3 were 1.55 A mg $_{Pt}{}^{-1}$ and 0.93 A mg $_{Pt+Pd}{}^{-1}$, 0.75 A mg $_{Pt}{}^{-1}$ and 0.48 A mg $_{Pt+Pd}{}^{-1}$, and 0.70 A mg$_{Pt}{}^{-1}$ and 0.4 A mg $_{Pt+Pd}{}^{-1}$ (Table S5). The Pd-Pt-Ni Octa1 had 9.68 ($M_{Pt}$) and 5.81 times ($M_{Pt+Pd}$) of Pt/C (0.16 mg $_{Pt}{}^{-1}$); Pd-Pt-Ni Octa2 had 4.68 ($M_{Pt}$) and 3 times ($M_{Pt+Pd}$) of Pt/C (0.16 mg $_{Pt}{}^{-1}$); Pd-Pt-Ni Octa3 had 4.37 ($M_{Pt}$) and 2.5 times ($M_{Pt+Pd}$) of Pt/C (0.16 mg $_{Pt}{}^{-1}$). ECSA refers to the active area of precious metals per unit mass. The specific activity (SA) refers to the normalization of the catalytic activity to the active material load or the active area of the catalyst; it represents intrinsic activity. The SAs of the Pd-Pt-Ni Octa1 were 1.87 mA cm$^{-2}$ $_{Pt+Pd}$, of the Pd-Pt-Ni Octa2 were 0.51 mA cm$^{-2}$ $_{Pt+Pd}$, and of the Pd-Pt-Ni Octa3 were 0.42 mA cm$^{-2}$ $_{Pt+Pd}$. It was 6.55, 1.72, and 1.41 times those of commercial Pt/C (0.29 mA cm$^{-2}$).

The durability of the Pd-Pt-Ni Octa1 was tested by 10,000 cycles in 0.1 M $HClO_4$. It was obvious that Pd-Pt-Ni Octa1 still had a relatively stable structure (Figure S4). As shown in Figure 6d, the half-wave potential of Pd-Pt-Ni Octa1 decreased by 17 mV after the stability test of 10,000 cycles, and the MA of Pd-Pt-Ni Octa1 was 0.51 A mg $_{Pt+Pd}{}^{-1}$ (Figure 6h). The half-wave potential of commercial Pt/C was reduced by 16 mV after the stability test of 10,000 cycles, and the MA of Pt/C was 0.09 A mg $_{Pt+Pd}{}^{-1}$ (Figure 6h, Table S6). It can be seen that Pd-Pt-Ni Octa1 exhibits a significant advantage over Pt/C in stability.

According to the electrochemical properties, it was obvious that the ORR activity of Pd-Pt-Ni Octa1 (4.71 nm) was higher than the other two particle size catalysts, which indicates that the ORR activity was also affected by the larger compression strain. The ultra-small Pd-Pt-Ni Octa1 had the highest lattice compressibility rate and exhibited the best ORR mass activity.

### 2.3. Theoretical Calculations

To further study the alloying effect of Pt with Pd and Ni elements on the ORR, the first-principles density functional theory (DFT) calculations for the bulk states of Pt/C and Pt-Pd-Ni trimetallic catalysts using a VASP [25,39–42] structural model diagram are as shown in Figure 7a,b, and the crystal planes (top views) are shown in Figure 7c,d.

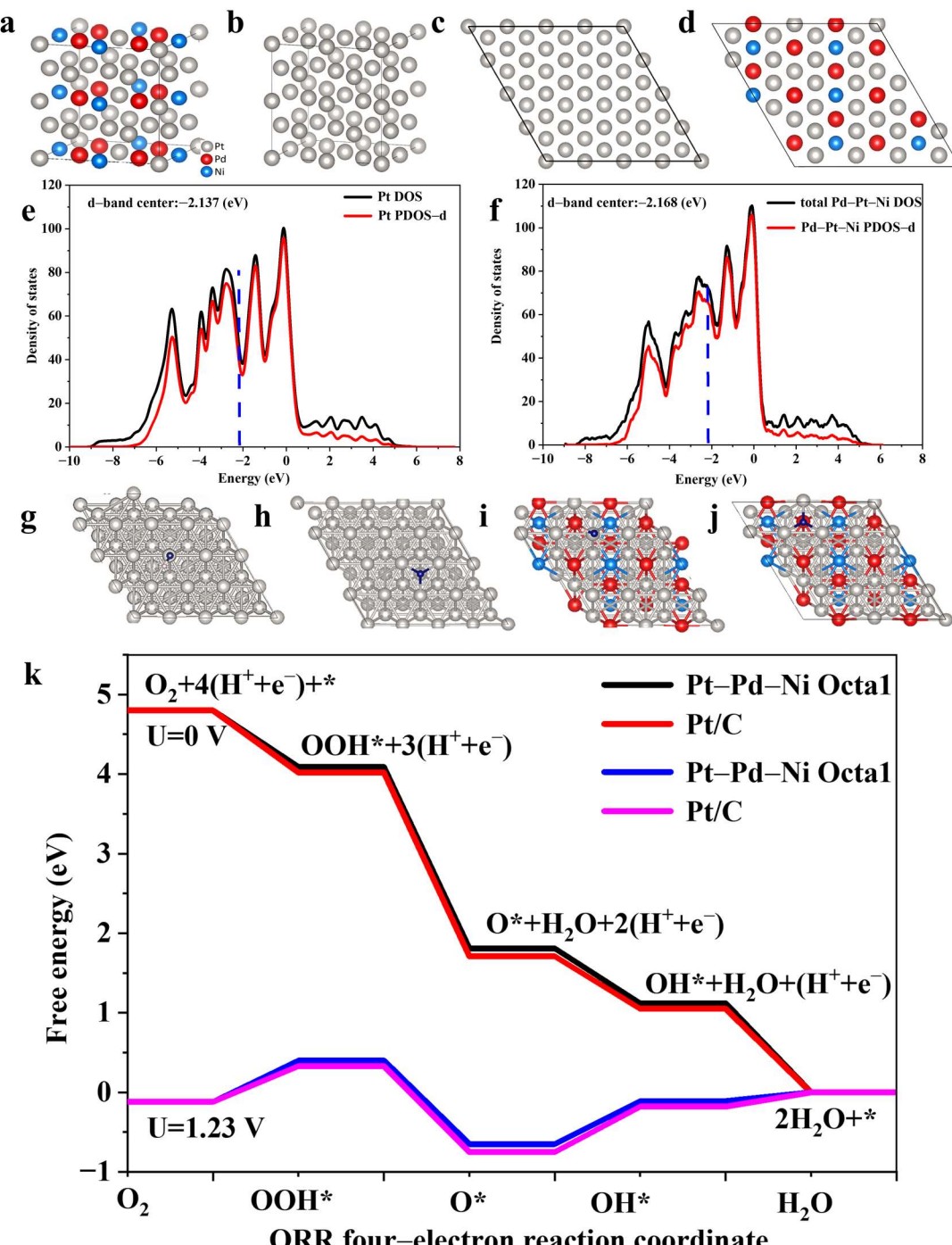

**Figure 7.** (**a**) Volume model diagram of the Pt/C catalyst; (**b**) volume model diagram of the Pt-Pd-Ni ternary nanocatalyst; (**c**) schematic diagram of the crystal plane of the Pt/C catalyst (111); (**d**) schematic diagram of the crystal plane structure of the Pt-Pd-Ni ternary nanocatalyst (111); (**e**) Pt/C state density diagram; (**f**) State density diagram of the Pt-Pd-Ni ternary nanocatalyst; (**g**) adsorption structure of the intermediate O* on the Pt (111) surface; (**h**) adsorption configuration of the intermediate OH* Pt (111) surface; (**i**) adsorption structure of the intermediate O* on the Pt-Pd-Ni Octa1 (111) surface; (**j**) adsorption configuration on the surface of the intermediate OH* Pt-Pd-Ni Octa1 (111); (**k**) Gibbs free energy changes at each step of the four-electron ORR.

It was found that the distance between the Pt atoms and other atoms in the model of the Pd-Pt-Ni (111) surface on Pd-Pt-Ni Octa1 were significantly shorter than that of the Pt's (111) surface (as shown in Table S2), which is consistent with the conclusions

obtained by the XRD and HRTEM characterization in the experiment. In general, the lattice compressibility on the catalytic surface reduces the distance of adjacent metal atoms on the crystal surface, increasing the d-band broadening [41]. However, we know that electrons tend to preferentially distribute in the orbital with lower energy according to the properties of electrons. Therefore, the d-band center of the Pd-Pt-Ni electrocatalysts tends to decrease relative to the Fermi level as well [25,39,40]. According to Xu et al., the lower d-band center can activate adsorbed oxygen and oxygen-containing intermediates in the bonding stage of the oxygen reduction reaction, which is conducive to the progress of ORR [43].

To directly compare the influence of the Pd-Pt-Ni Octa1 of the Pt-Pd-Ni ternary alloy on the Pt d-band center ($\varepsilon_d$), the state density of the Pt-Pd-Ni (111) surface on the Pt-Pd-Ni and Pt/C was calculated, as shown in the state density diagram in Figure 7e,f. The d-band center of Pt on the Pt (111) crystal plane was $-2.137$ eV, while the plane was $-2.168$ eV. As previously predicted, the d-band center of the Pt-Pd-Ni (111) shifts negatively concerning the surface of pure Pt (111) after the alloy was formed with Pd and Ni. According to the d-band center theory, the d-band center shifted negatively, which indicates that the contraction strain occurred on the crystal surface, and the interaction of the Pt atomic adsorbent was weakened. Therefore, under the interaction influence between the Ni and Pd metals and Pt, the electronic structure of Pt could essentially be adjusted, so that the center of the d-band could be adjusted to optimize the adsorption performance and enhance the activity.

In addition, the Gibbs free energy of Pt-Pd-Ni Octa1 was calculated (Figure 7g–j). As shown in Figure 7k, the ORR process on the (111) surface of the Pt and Pt-Pd-Ni was spontaneous, which was analyzed from the perspective of thermodynamics [40,44,45]. However, some studies have shown that the process of OH* desorption is a key step that limits the reaction rate of oxygen reduction on the surface of Pt-based catalysts [15,39]. It is obvious that promoting the dissociation of the O-O bond within *OOH is key to the four-electron reaction path. In many steps in the four electron reaction, the rate determination step (RDS) based on ORR of binding energy was established. In the calculation process, there was a certain linear relationship between the intermediate products produced in each step: $\Delta G_O = 2\Delta G_{OH}$ and $\Delta G_{OOH} = \Delta G_{OH} + 3.2$. Thus, $\Delta G_{OH}$ can evaluate the ORR activity by single molecule binding free energy. In the calculations process, the Gibbs free energy for ORR in Figure 7k shows that the OH* desorption on Pt-Pd-Ni Octa1 surface exhibited the lowest overpotential of 0.54 eV, while the Pt/C was 0.57 eV (Tables S3–S5). The results indicate that the ultra-small octahedra formed by depositing Pt-Ni on Pd nanocube seed had a strong alloying effect, which strongly affected the ORR activity.

## 3. Experimental

### 3.1. Materials

Chloroplatinic acid hexahydrate ($H_2PtCl_6 \cdot 6H_2O$, 4.8 mg/mL), palladium chloride ($PdCl_2$, 98%), nickel (II) acetylacetonate ($Ni(acac)_2$, 95%), tungsten hexacarbonyl ($W(CO)_6$), oleylamine (OAM, 80–90%), oleic acid (OA), benzyl alcohol (BA, 99.8%), poly(vinylpyrrolidone) (PVP), potassium bromide (KBr, 99%), and deionized (DI) water.

### 3.2. Synthesis of Pd NCs

First, 105 mg PVP, 60 mg AA, and 400 mg KBr were added into 8 mL DI water and stirred at 80 °C for 20 min until dissolved into a transparent form and, subsequently, slowly added to 3 mL low solubility solution containing 57 mg Pdcl$_2$ with a dropper and stirred at room temperature for 30 min under the assistance of ultrasound. When the two were fully mixed, the color became dark red. Finally, Pd nanocube seeds were synthesized by heating them in an oil bath at 80 °C for 3 h. The obtained products were washed several times using acetone and ethanol, the mixture was centrifuged at 10,000 rpm to obtain the products after cooling to 25 °C.

Second, regarding the treatment of the transition from the DI water system to the OAM system, we added a total of 11 mL OAM into centrifuged Pd nanocube seeds in batches,

mixed them with ultrasonic assistance, transferred them to glass bottles, and put them into oil bath for heating. The temperature was raised from room temperature to 180 °C, and the mixture was stirred for 40 min. Note that in this step, the lid of the glass bottle was opened to remove the residual water contained in the Pd nanocubes seed, which used the evaporation principle. At this time, the solvent in the reaction was only OAM.

Finally, we closed the lid and then let react for 3 h. In the process of the Pt-Ni deposition, we added 2 mL BA to play the role of dispersion.

### 3.3. Synthesis of Pd-Pt-Ni Octa1

The above synthesized Pd NCs were kept at 80 °C for 1 h with stirring in an oil bath. At the same time, 815 μL of $H_2PtCl_6\cdot$ and 11 mg of $Ni(acac)_2$ were added into a 50 mL glass bottle. Further, 5 mL OAM, 1 mL OA, and 2 mL BA were injected into the mixture, and then stirred for an hour at room temperature. Then, 2 mL of Pd NCs seeds were transferred into the above solution, and the temperature of the mixture was kept at 130 °C for 1 h with stirring. Finally, the mixture and 50 mg of $W(CO)_6$ were added into a 25 mL reactor liner and heated to 200 °C for 6 h. For the synthesis of Pd-Pt-Ni **Octa2**, the preparation process was the same as for the synthesis of Pd-Pt-Ni Octa1, except that the amount of Ni was increased to 17 mg, while for the synthesis of Pd-Pt-Ni **Octa3**, the preparation process was the same as the synthesis of Pd-Pt-Ni Octa2, except that the amount of $H_2PtCl_6$ was decreased to 407 μL.

### 3.4. Preparation of Carbon-Supported Catalyst

The synthesized catalysts were dried, then mixed with n-butylamine and treated with ultrasound for 1 h. At the same time, 30 mg of commercial carbon black was mixed with 15 mL of N-butylamine and treated with ultrasound for 1 h. Subsequently, the two solutions were mixed, and the mixture was sonicated for 2 h. Then, the ultrasonic solution was stirred magnetically for three days and centrifuged for 15 min. Finally, the products were washed several times with ethanol and methanol and then put into the vacuum oven to dry at 70 °C for 12 h.

### 3.5. Electrochemical Characterization

In electrochemical testing, we used a three-electrode system of rotating disk electrodes (RDE) and a biologic VSP electrochemical workstation for testing. A reversible hydrogen electrode (RHE) was used as the reference electrode, and a platinum plate was used as the s counter electrode. The 3 mg Pt/C catalyst (TKK, 46.7 wt% Pt) was mixed with 1.5 mL ethanol and 20 μL of 0.5 wt% Nafion solution, and ultrasound was performed for 1 h. Then, 10 μL of the slurry was dropped onto the disk electrode and let dry [38]. Next, 2.7 mg of different carbon-supported Pd-Pt-Ni Octa1, Pd-Pt-Ni Octa2, and Pd-Pt-Ni Octa3 were mixed in 15 μL of 0.5 wt% Nafion solution and 1.2 mL of ethanol. Then, the ink was dropped onto the disk electrode according to the above steps, making it a uniform film formation.

In the electrochemical test, 0.1 M $HClO_4$ solution was used as the electrolyte solution. Activation (scanning rate: approximately 100 mV/s) and collection cycle voltammetry curves (CV) (scanning rate: approximately 0.05 V/s) were performed under a saturated $N_2$ environment. Under the condition of oxygen saturation, the disk electrode was rotated (rotation rate was approximately 1600 rpm), and finally the linear scanning voltammetry curve (LSV) was collected. The scanning range of the above tests was approximately 0.05–1.1 V versus RHE. Regarding the final durability test (ADT), 10,000 cycles (scanning rate: approximately 100 mV/s; range: approximately 0.6–1.0 V) were carried out under oxygen saturation conditions, and finally the LSV was tested.

### 3.6. Theoretical Calculations Details

All of the first principle calculations were performed using the Vienna Ab initio simulation Software Package (VASP), and the potential field of the valence electrons was

described using the projection-add plane wave (PAW) method. In the calculation process, the k-point grid was set as $13 \times 13 \times 13$ and $5 \times 5 \times 1$ respectively, when the structures of bulk states and crystal surface structures were optimized. In the process of the structural optimization, the Hellmann–Feynman force applied to each atom was gradually reduced to $10^{-2}$ eV $\text{Å}^{-1}$, and the energy difference obtained from two adjacent self-consistencies was gradually reduced to $10^{-6}$ eV [4,34–36,38–40]. The cut-off energy of the plane wave was set to 500 eV. Double vacuum layers with 15 Å were added to isolate the periodic influence in the vertical direction of the surface. The equilibrium lattice constants of the Pt unit cell were optimized, when using a $13 \times 13 \times 13$ Monkhorst–Pack k-point grid for Brillouin zone sampling. These lattice constants were used to build the p($4 \times 4$) Pt(111) surface with 4 atomic layers, which contained 64 Pt atoms, and to build the PtPdNi surface, including 40 Pt atoms, 16 Pd atoms, and 8 Ni atoms. This slab was separated by a 15 Å vacuum layer in the z direction between the slab and its periodic images. During structural optimizations of the surface models, a $5 \times 5 \times 1$ gamma-point centered k-point grid for the Brillouin zone was used, and the top 2 atomic layers were allowed to fully relax, while the bottom atomic 2 layers were fixed.

Among them, the Gibbs's free energy change was calculated by the following formula: $\Delta G_{ads} = \Delta E_{ads} + \Delta ZPE - T\Delta S$; $\Delta G = \Delta G_{ads} + \Delta G_{species}$, where ads represents the intermediate species in the state of adsorption, $E_{ads}$ represents the adsorption energy, $\Delta ZPE$ represents the zero-point energy change of the two adsorption states before and after, $T\Delta S$ represents the change of entropy in the reaction process, and $\Delta G_{species}$ represents the difference in the Gibbs free energy of the desorption product in the initial and final states of the reaction. The adsorption and zero-point energies of the various intermediates can be calculated by the first principles. The entropy change can also be obtained using VaspKit. At this point, the change in the Gibb's free energy at each step can be expressed by the following equations: $\Delta G_1 = \Delta G(*OOH) - 4.8$, $\Delta G_2 = \Delta G(*O) - \Delta G(*OOH)$, $\Delta G_3 = \Delta G(*OH) - \Delta G(*O)$, and $\Delta G_4 = \Delta G(H_2O) - \Delta G(*OH)$.

## 4. Conclusions

Ultra-small nanoparticles of Pt-Ni-Pd ternary alloy octahedra were successfully synthesized through the Pd-seed-inducing growth route using Pd nanocubes as key intermediates. The small-sized ternary alloy octahedra (Pd-Pt Ni octa1, 4.71 nm) were obtained based on the deposition of Pt-Ni to induce Pd seeds for lattice distortion and structure remodeling. The Pd seeds were oxidized into $Pd^{2+}$ and combined with $Pt^{2+}$ and $Ni^{2+}$ in the solution, finally forming the ternary alloy small-sized octahedra. The feeding ratio of Pt-Ni in the mixed deposition reaction strongly affected the structural changes and lattice distortion of the Pd seeds, which is reflected in the size changes of different octahedra and the different lattice mismatch rates of the Pd/C (111) crystal plane and Pd-Pt-Ni (111) crystal plane. This showed different ORR performances. The ultra-small Pd-Pt Ni Octa1 was synthesized with a feeding ratio of 2:1, showing a high contraction strain, and exhibited excellent ORR performances. Its half-wave potential was 0.933 V in the acidic solution. Meanwhile, the theoretical calculation showed that the Pt-Pd-Ni (111) crystal of the Pd-Pt-Ni Octa1 had a more significant lattice contraction (contraction rate: 3.49%), and the d-band center showed an obvious negative shift. In the four-electron reaction, the OH* desorption on the Pt-Pd-Ni Octa1 surface exhibited the lowest overpotential, resulting in the adsorption performance being optimized.

Most importantly, this route combines the enhancement of the catalytic active site with the optimization of the Pt electronic structure and the synthetic method to prepare Pt-based small-sized catalysts for ternary alloy. The Pd seed-inducing-growth route provides a feasible strategy to develop highly active catalysts through the selection of many 3d metals with strong synergistic effects on precious metals ($Co^{2+}$ and $Cu^{2+}$) and precious metals seed with an electronic structure similar to Pt (Ir, Rh, etc.). Therefore, it is reliable that the route we proposed provides a new idea for exploring the synthesis of small-sized structures for ORR and other electro-catalytic reactions.

**Supplementary Materials:** The following supporting information can be downloaded at: https://www.mdpi.com/article/10.3390/catal13010097/s1. Figure S1. TEM image of Pd-Pt-Ni Octa1 and responding particle size distribution; Figure S2. TEM image of Pd-Pt-Ni Octa2 and responding particle size distribution; Figure S3. TEM image of Pd-Pt-Ni Octa3 and responding particle size distribution; Figure S4. TEM images after 10 k cycles of Pd-Pt-Ni Octa1. Table S1. Pt, Pd and Ni atomic ratio examined of Pd-Pt-Ni Octa1, Pd-Pt-Ni Octa2, Pd-Pt-Ni Octa3 by XPS and ICP-OES; Table S2. The relevant Lattice parameter correlated to Pt(111) plane (nm) and calculated strain of Pt, Pd-Pt-Ni Octa1, Pd-Pt-Ni Octa2 and Pd-Pt-Ni Octa3; Table S3. When the gas phase water was at 0.035 bar and T = 298 K, the gas phase water was in equilibrium with liquid water. The following data are calculated at this equilibrium state; Table S4. Gibbs free energy in a four-electron reaction, while product1 stands for 2 ($H^+ + e^-$) + $H_2O$ and product2 stands for $H^+ + e^-$ + $H_2O$, U = 0 V; Table S5. In the four-electron reaction, the Gipps free energy of Pd-Pt-Ni Octa1 (111) in ORR each step at 1.23V is as follows: $\Delta G(H_2O) = 0$, $\Delta G(*OH) = 1.12–1.23$, $\Delta G(*O) = 1.81–2*1.23$, $\Delta G(*OOH) = 4.09–3*1.23$, $\Delta G(O_2)$ = 4.8–4*1.23. In the four-electron reaction, the Gipps free energy of Pt/C in ORR each step at 1.23V is as follows: $\Delta G(H_2O) = 0$, $\Delta G(*OH) = 1.05–1.23$, $\Delta G(*O) = 1.71–2*1.23$, $\Delta G(*OOH) = 4.02–3*1.23$, $\Delta G(O_2) = 4.8–4*1.23$; Table S6. Performance of Pd-Pt-Ni Octa1, Pd-Pt-Ni Octa2 and Pd-Pt-Ni Octa3 catalyst compared to Pt/C.

**Author Contributions:** Investigation, Data Curation, Validation, Writing—Original Draft, Methodology, Y.L., W.L. and H.F.; Data Curation, Validation, and Methodology, Z.L., Q.C., G.L. and X.H.; Conceptualization, Writing—Review and Editing, P.T.; Supervision, Project Administration, Resources, and Funding Acquisition, P.S. All authors have read and agreed to the published version of the manuscript.

**Funding:** This work was funded by the National Natural Science Foundation of China (22075055) and the Guangxi Science and Technology Project (AB16380030).

**Data Availability Statement:** Not applicable.

**Conflicts of Interest:** The authors declare no conflict of interest.

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
