# Peer review of "Ultra-Small Nanoparticles of Pd-Pt-Ni Alloy Octahedra with High Lattice Strain for Efficient Oxygen Reduction Reaction"

_catalysts, doi:10.3390/catal13010097_

Round 1
Reviewer 1 Report
The work reports a synthesis of small-size Pd-Pd-Ni octahedra via a seed-mediated growth for oxygen reduction reaction with improved activity. The catalyst was systematically analyzed to study the size, surface facets, and compositions. The roles played by Pd seeds in the formation of the ternary octahedra was also investigated and presented. However, there are many technical and expression issues in this work, which have to be addressed for further consideration of acceptance.
Comments:
1) There is no reasonable explanation for the occurrence of galvanic replacement reaction between Pd seeds and the Pt ions in the deposition process. Based on my knowledge, the seed-mediated growth of Pt-Ni overlayers on Pd seeds have been applied using Pd octahedral seeds via the same protocol (ACS Nano 2014, 8, 10, 10363–10371). However, although the reduction potential of Pt is higher than that of Pd, no remarkable galvanic replacement reaction was reported when using Pd as seeds for the overgrowth of Pt. Therefore, phenomenon should be well explained under the reaction condition.
2) There are two major concerns for me in the experimental section. The first one is the dissolution of PdCl2 in DI water. Considering the ultralow solubility of PdCl2 in water under neutral condition, I am quite doubtful if 60 mg of PdCl2 can either be dissolved or dispersed in 3 mL water even by ultrasonication for 5 min. The second one is the dispersion of Pd seeds in OAm. Since the Pd seeds were synthesized in a hydrophilic solvent, the direct dispersion of them in the hydrophobic solvent such as OAm will be very hard, which requires an amphiphilic solvent such as benzyl alcohol (ACS Nano 2014, 8, 10, 10363–10371). Therefore, the description of dispersion of Pd seeds in 11 mL OAm is not reasonable to me.
3) In Figure 2c, the formation of large octahedra should be further explained. The products were obtained by disposition of Ni on the Pd seeds with a size of 9 nm, which is not understandable to me for the generation of microsize octahedra.
4) The language of the manuscript is poor with many reasonable descriptions and terms, to which the author needs to pay more attention to make it eligible for publication. Here are some examples:
· Page 1, line 27, “concentration ratio” should be “feeding ratio”
· Page 1, line 29, “octahedra1” should be “octahedral”
· Page 1, line 29, “Lattice compressibility rate” should be “Lattice contraction”
· Page 2, line 55, “larger catalyst” should be “catalyst with larger sizes”
· Page 4, line 179, “According to the volcano diagram theory presented by Stamenkovic et al., we can know that Pt atoms have lower coordination numbers or more dangling bonds than transition metal atoms”, line 183, “Nickel (Ni) is the best transition metal element.”. These descriptions do not make sense to me.
· Page 4, line 179, “oleoamine” should be “oleylamine”
Reviewer 2 Report
In this work, the authors report a Pd seed inducing-growth route to synthesize Pd-Pt-Ni alloy octahedra, its ultra-small nanoparticles together with high lattice strain making it exhibit promising catalytic activity for oxygen reduction reaction (ORR). The authors proposed that this efficient route not only increases the number of catalytic active sites, but also optimizes the Pt electronic structure through alloying effect affect. DFT calculations pointed out that the d-band center of Pt species negatively shift in Pd-Pt-Ni (111) structure, further weakened the interaction of Pt atomic adsorbent. The data is sufficient, but I think the structure of the manuscript need to be further improved. Therefore, I recommend a major revision for this work before publication in Catalysts. Here are some issues which should be carefully addressed.
1. I noticed that in XPS the binding energy of Pt and Pd exhibit obviously positive shift for Pd-Pt-Ni Octa1, Pd-Pt-Ni Octa2 and Pd-Pt-Ni Octa3, could this result be further support by DFT results?
2. For DFT calculations, it is well known that the theoretical models make great contributions to the calculation results, which greatly influence the final conclusion of the whole system. However in this work, the author did not give any details of model information, which I think it is very difficult for other researchers to repeat this work, so I recommend the authors adding necessary details for their theoretical models.
3. For catalytic stability, more characterization results should be added in the manuscript.
4. As for intrinsic activity, it is recommended adding the ECSA-normalized catalytic activities in the work.
5. The qualities of figures and logical structure need to be improved.
Reviewer 3 Report
The manuscript with title Ultra-small nanoparticles of Pd-Pt-Ni alloy octahedra with high 2 lattice strain for efficient oxygen reduction reaction is very interesting and good written. The presentation is very good. I recommended it after minor changes.
1. Need to revise Abstract in numerical form.
2. There is need to revised result and discussion with latest literature.
3. Why author use Pd-Pt-Ni alloy octahedra?
4. Pd-Pt-Ni alloy octahedra with high 2 lattice strain. why not author test on low intensity strain.
5. cite the following literature.
Round 2
Reviewer 1 Report
The authors have addressed the some parts of the technical and language issues. And the manuscript can be accepted after careful proof.
Reviewer 2 Report
The authors' answers fully address my concerns, therefore I recommend publishing this work in Catalysts.